# Rabies control via co-creation: A model for sustainable one health interventions

Nicholas Bor[1,2], Geoffrey Njenga[1], Annabel Slater[1], Peterkin Munywoki[1],
Dorcas Chepyatich[1], David Owino[3], Maurice K. Murungi[1,4], Dishon M. Muloi[1,4],
Lian F. Thomas[1,2,4]*

1 International Livestock Research Institute, Nairobi, Kenya, 2 Royal (Dick) School of Veterinary Studies, University of Edinburgh, Midlothian, United Kingdom, 3 Vétérinaires Sans Frontières Germany, Nairobi, Kenya, 4 Institute of Infection and Global Health, University of Liverpool, Leahurst Campus, Neston, United Kingdom

* N.K.Bor@sms.ed.ac.uk

## Abstract

Rabies is a fatal zoonotic disease that can be effectively controlled by vaccinating 70% of at-risk dog populations. Rabies is a persistent health threat in Kenya due to low rabies vaccination coverage and low awareness among dog owners. In 2014, the Zoonotic Disease Unit launched the national rabies elimination strategy and listed Machakos County as one of the pilot counties due to the high rabies burden. Between 2021 and 2024, the International Livestock Research Institute, Veterinarians Sans Frontières - Germany and Machakos County government implemented a series of free mass dog rabies vaccination campaigns. To support vaccine uptake, the research team co-created contextual rabies sensitization materials and vaccination announcement posters with the community. This was achieved through a series of key informant interviews, focus group discussions and co-creation workshops with community members who presented their dogs for vaccination and agreed to be part of these activities. Community members shared that dogs play both tangible and intangible roles in their lives. The main tangible roles included guarding homes, livestock, and crops. Friendship was highlighted as the most important intangible role. They stressed that a bond of friendship must first be established for dogs to serve their owners meaningfully. They shared that the co-creation process deepened their understanding of rabies and helped them better appreciate the value of their dogs. As a result, they recognized the importance of vaccinating their dogs — their friends — against the disease. Co-creating public health solutions with communities is an invaluable approach to raising awareness, building trust, and boosting participation in campaigns, all of which are vital for long-term sustainability. We advocate for greater use of co-creation processes, especially as we apply the One Health approach to address zoonotic threats and other health challenges.

**Data availability statement:** Data is available on this link: https://data.mel.cgiar.org/dataset.xhtml?persistentId=hdl:20.500.11766.1/FK2/Y3KBBT.

**Funding:** This work was supported by the German Federal Ministry for Economic Cooperation and Development (BMZ) which was sent to the One Health Research, Education and Outreach Centre in Africa (OHRECA) based at ILRI. OHRECA also received the Participatory Research Grants Scheme from the University of Liverpool. The funders did not play any role in study design, data collection and analysis, decision to publish, or preparation of the manuscript.

**Competing interests:** The authors have declared that no competing interests exist.

## Author summary

Rabies remains endemic in Kenya due to low vaccination coverage and limited public awareness. A mass dog rabies vaccination campaign was implemented in Machakos County between 2021 and 2024, but coverage has remained sub-optimal. To help increase vaccine uptake, we co-created vaccination announcement posters and educational materials with community members. We began by conducting key informant interviews and focus group discussions with dog owners to understand the value they place on their dogs. These insights then informed co-creation workshops where the materials were developed collaboratively. Our findings highlight that dogs serve both tangible and intangible roles in the community. Tangible roles included protecting homes, livestock, and crops, while companionship was identified as the most important intangible role. Notably, participants viewed friendship as a prerequisite for dogs to provide meaningful service. The co-creation process allowed community members to reflect on the value of dogs, improved awareness of rabies, and increased motivation to vaccinate. We recommend incorporating co-creation approaches into public health interventions, as they build trust, enhance awareness, and support community engagement—factors critical to the sustainability of One Health initiatives.

## Introduction

Rabies is a viral zoonotic disease caused by a *lyssavirus* and affects all mammals [1]. Domestic dogs are responsible for 99% of human rabies cases where transmission occurs through bites and scratches from rabid dogs [2]. Globally, rabies causes 59,000 human deaths annually [3] with approximately 41% of these deaths occurring in Africa [4]. While rabies has been eliminated in most high-income countries, it persists in low and middle-income countries, leading the World Health Organization (WHO) to classify it as a neglected tropical disease [5]. A 2005–2018 review of rabies incidents and mortalities in 28 African countries showed that Kenya reported the highest number of canine rabies cases to the World Organization for Animal Health (WOAH) database [6].

Rabies is one of the top five priority zoonotic diseases in Kenya with human and animal health professionals advocating for increased resource allocation towards its elimination [7]. In response, the government through the Zoonotic Disease Unit (ZDU), which is mandated to coordinate and implement the One Health approach to managing zoonotic diseases, developed the National Rabies Elimination Strategy (2014–2030). The strategy aims to eliminate dog-mediated rabies by 2030, with Machakos County selected as one of the pilot counties due to its high rabies burden [8].

Rabies control strategies focus on vaccinating 70% of at-risk dog populations and timely administration of post-exposure prophylaxis (PEP) to dog bite and scratch victims [1]. Despite these seemingly straightforward control strategies, barriers to rabies

elimination exist in different settings. These barriers include resource constraints, limited collaboration, and low rabies awareness among communities [1,9]. Sensitization, an act of raising awareness on a particular subject, has been shown to increase rabies vaccine uptake and encourage participation as evidenced in different communities in Burkina Faso and Tanzania [10,11].

The One Health High Level Expert Panel, an advisory group to the quadripartite organizations, supports community participation as a fundamental aspect of global health policies and disease control strategies. It emphasizes inclusivity, equity, and accessibility [12].

Co-creation is an example of community participation method where community members, researchers, and policymakers unite to address public health challenges by leveraging the community's insights and scientific expertise in designing and delivering health interventions [13]. By co-creating health solutions with communities, involved stakeholders build trust and foster knowledge sharing for better health outcomes [14]. Researchers and policy makers acknowledge the community's perspectives and stimulate productive engagements, thus making communities active producers and disseminators of campaign messages [15].

As a step towards supporting community engagement with rabies control, this study sought to understand the value of dogs to the owners and co-create rabies sensitization materials that incorporate these values.

## Methods

### Ethical approval and participants' consent

ILRI's Institutional Research Ethics Committee (ILRI-IREC 2023-09) and the National Commission for Science, Technology, and Innovation (NACOSTI/P/23/25371) approved this study. Written consents were obtained from all study participants while community members who co-created sensitization materials have been acknowledged. This study also received approval from the University of Liverpool Research Ethics Committee reference number 12560.

This study involved collecting data on the value of dogs as described by the community, gathering ideas on ideal rabies sensitization materials, and obtaining feedback on the draft materials and research experiences.

### Study site

In October 2021, the International Livestock Research Institute (ILRI), Vétérinaires Sans Frontières Germany (VSF-Germany), and the Machakos County Government initiated a free rabies vaccination campaign in Machakos County. The study was conducted in Mwala sub-county where we capitalized on participant recruitment during the ongoing rabies vaccination campaign. Mwala sub-county has a human population of 181,896 [16]. This semi-arid region primarily relies on rain-fed agriculture and small-scale livestock keeping. The residents cultivate maize, beans, fruits, vegetables, sorghum, and millet. Commonly reared livestock species include goats, sheep, dairy and beef cattle [16].

### Key informant interviews

Community members who presented their dogs for the vaccination campaign were conveniently recruited to the study. Informed consent was obtained from each participant, and a semi-structured interview lasting approximately five minutes was conducted. These interviews aimed to identify the value of dogs to the community with the goal of co-creating relevant sensitization materials that align with these values. The interviews were held on 22 – 24 June 2023. Two enumerators were responsible for conducting an average of 34 interviews daily.

The interview guide for the KIIs was developed in English. However, interviews were conducted in the language most comfortable for each respondent: either English, Kiswahili, or Kikamba. Two enumerators (NB and PM) translated the guide into the respondent's preferred language. Enumerator NB was fluent in English and Kiswahili, while PM was fluent in all the three languages. Respondents who preferred to be interviewed in Kikamba were assigned to PM.

The subsequent research activities – FGD and co-creation workshops— were conducted in English and Kiswahili as the randomly selected participants reported being comfortable with both languages. These sessions were similarly guided by tools developed in English, with real-time translation by the facilitators into the language preferred by participants. This approach ensured participants could express their views with clarity.

## Data transcription and analysis

The interview audio responses were recorded and later transcribed verbatim in Microsoft Word. All the transcripts from the KIIs were reviewed to identified the value of dogs to the owners and the wider community. Reading and reviewing the transcripts allowed us to inductively identify and summarize the value of dogs [17].

## Focus group discussion (FGD)

All key informants were assigned unique numbers. Twenty-four key informants were randomly selected using a random number generator [18]. The selected community members were invited for two gender-segregated FGDs. We presented the summarized roles and values of the dogs to FGD attendees. Each value was displayed on a sticky note and explained to them. Pairwise ranking [19] was conducted collectively with the participants and the dog values were ranked from the most important to the least important. Different pairs of dog values were compared against each other from the summarized dog values. The participants would choose the 'more important' dog value against each pair. For example, security was compared against societal norm and if security was perceived to be more important than societal norm, the assigned letter for security would be recorded on that cell. The assigned value that 'won' against each pair was counted across that row to come up with a score. The scores were used to ranking the value of dogs as shown in Tables 1 and 2.

At the end of the FGD, the same attendees were shown a rabies educational video produced by the ILRI team [20].

## Co-creation workshops

All FGD participants were invited to participate in two co-creation workshops. In the first workshop, they worked in groups to co-create rabies sensitization materials. Products from this session were used by the research team to develop prototypes of the sensitization materials that were then presented to the participants in the second co-creation workshop.

The aim of the second co-creation workshop was to identify the most relevant prototype of the vaccination announcement poster and educational poster. This workshop gave the participants an opportunity to select the best images to include in the materials, the colours to use and the reasons for their choices. This provided the research team with a clear understanding of their reasoning, which guided the production of the final output of the materials.

Table 1. Outcome of the pairwise ranking among female participants.

| Value of dogs | Assigned Value of Dog | | | | | | | | | | Score | Rank |
|---|---|---|---|---|---|---|---|---|---|---|---|---|
| | A | B | C | D | E | F | G | I | H | J | | |
| A. Security | | A | A | A | A | A | G | A | A | A | 8 | 1 |
| B. Societal norm | | | C | D | E | F | G | H | I | B | 1 | 9 |
| C. Herding | | | | C | E | F | G | C | C | C | 5 | 5 |
| D. Hunting | | | | | E | F | G | H | I | J | 2 | 8 |
| E. Crop protection | | | | | | F | G | E | E | E | 6 | 4 |
| F. Livestock protection | | | | | | | F | F | F | F | 8 | 1 |
| G. Pet and Friend | | | | | | | | G | G | G | 8 | 1 |
| H. Status symbol | | | | | | | | | I | H | 3 | 7 |
| I. Source of income through breeding | | | | | | | | | | I | 4 | 6 |
| J. Give away puppies | | | | | | | | | | | 0 | 10 |

**Table 2. Outcome of the pairwise ranking among male participants.**

| Value of dogs | Assigned Value of Dog | | | | | | | | | | Score | Rank |
|---|---|---|---|---|---|---|---|---|---|---|---|---|
| | A | B | C | D | E | F | G | H | I | J | | |
| A. Security | | A | A | A | A | A | A | A | A | A | 9 | 1 |
| B. Societal norm | | | B | B | E | F | G | B | B | B | 5 | 5 |
| C. Herding | | | | C | E | F | G | C | C | C | 4 | 6 |
| D. Hunting | | | | | E | F | G | H | D | D | 2 | 8 |
| E. Crop protection | | | | | | F | E | E | E | E | 7 | 3 |
| F. Livestock protection | | | | | | | F | F | F | F | 8 | 2 |
| G. Pet and Friend | | | | | | | | G | G | G | 6 | 4 |
| H. Status symbol | | | | | | | | | H | H | 3 | 7 |
| I. Source of income through breeding | | | | | | | | | | I | 1 | 9 |
| J. Give away puppies | | | | | | | | | | | 0 | 10 |

In both workshops, informed consent was obtained to photograph and acknowledge them as co-creators in the publication.

### 1st co-creation workshop: development of draft sensitization material by research participants

In the first co-creation workshop, the research team introduced the rabies topic and recapped the values of dogs as captured in the KIIs. This workshop was organized into ideation and storyboarding sessions. The ideation session harnessed the collective creativity of the participants to generate ideas, while the storyboarding session transformed the generated ideas into visual storyboards [21]. Each participant was given coloured sticky notes to write the exact words they wished to see on a rabies vaccination campaign poster (Fig 1). Participants were encouraged to withhold criticism while peers shared their ideas.

In the storyboarding session, participants translated their ideas into visual storyboards. Each group sketched their storyboards while focusing on the value of dogs in the community, their motivation for vaccinating them, and the feelings post vaccination. The completed storyboards were pinned on the walls for peer reviewing.

### Development of prototype sensitization materials by the research team

The first co-creation workshop revealed the need for two separate products: a rabies vaccination announcement poster and an educational poster. The research team transformed the co-created storyboards into two product prototypes. The rabies vaccination announcement poster was suggested for use during campaigns. The rabies educational poster would create awareness on rabies and the importance of vaccinating dogs against rabies.

### 2nd co-creation workshop: selection of images to incorporate into the sensitization materials

A strategic decision in our design process was to initially omit images from the prototype of the sensitization materials. This offered the participants an opportunity to select images that were relevant to their context.

The research team identified images that closely represent their context and the dog values. These images included photos taken during the vaccination campaigns while some were generated using via DALL.E 2 software - a text-to-image artificial intelligence image generator [22]. We used specific community context prompts and reference images to guide AI in generating the desired images. An example prompt was *"Generate an image that depicts the bond between individuals and their dogs, emphasizing companionship. The image should be contextualised to Kenya, Machakos County. The person should wear a yellow T-shirt".*

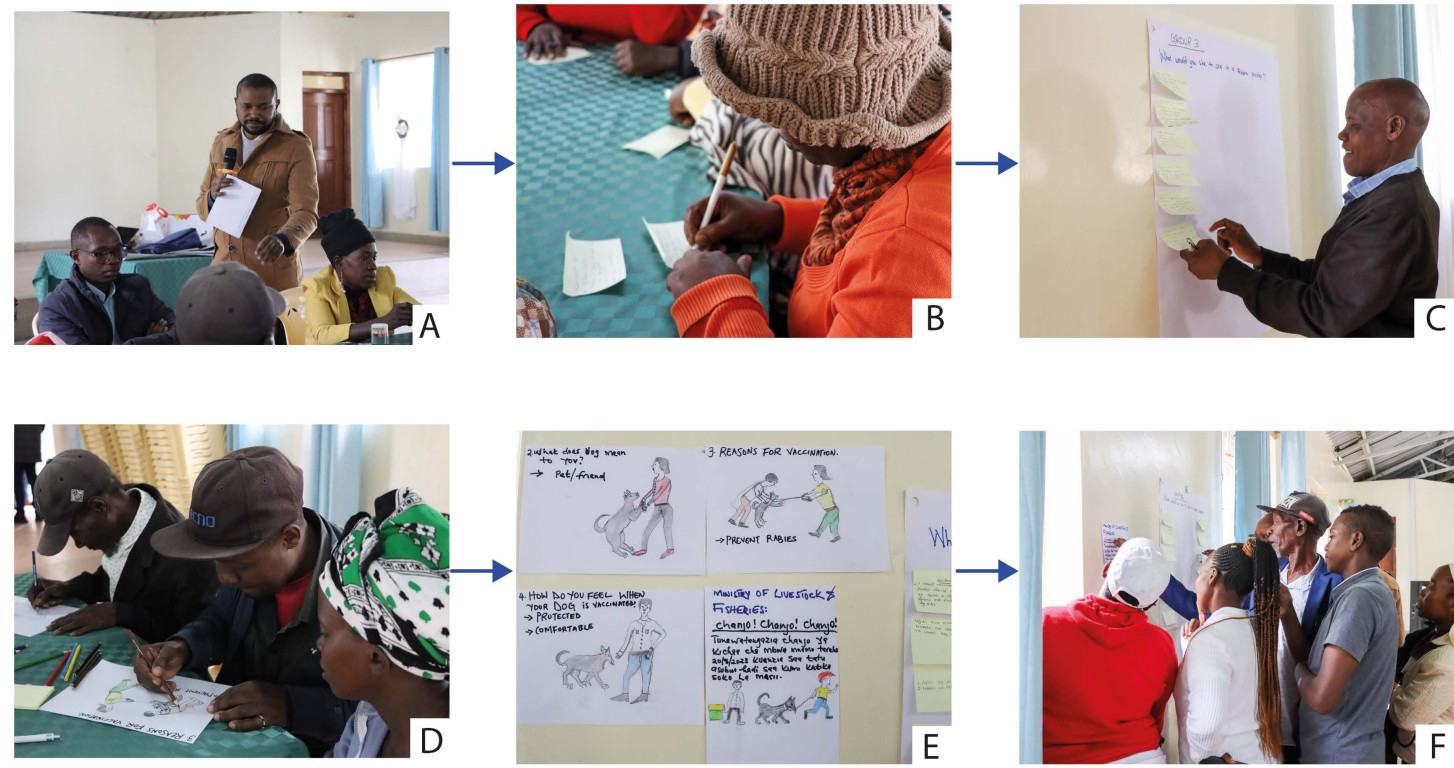

Photo credit: ILRI/Madeline Wong

**Fig 1. Flow diagram of the co-creation process to develop the initial drafts of the rabies materials.** A. Facilitator introduces the ideation and storyboarding processes to participants. B. Participant writes her ideas on the things she would like to see in a rabies poster. C. Participant pins his poster ideas on the wall. D. Group members transform their ideas into storyboards. E. Examples of completed storyboards. F. Participants review and critique the completed storyboards. They share what they like about the completed storyboards and give suggestions to improve the quality of the storyboards.

### Participants' perceptions of the research experience and co-creation processes

Finally, an exit survey was conducted to capture participants' experiences throughout the research and co-creation processes. They shared key learnings and provided suggestions for future interventions.

## Results

### Study participants

A total of 102 key informants, 68 males and 34 females, were initially recruited into the study. From these, twenty-four participants, 12 males and 12 females, were selected to participate in a focus group discussion and two co-creation workshops.

### Value of dogs to the community members

Interactions with community members revealed that dogs' roles also held significant value to them. In this paper, we use dogs' roles and values interchangeably. The roles were both tangible and intangible. Examples of tangible roles include providing security while intangible roles include providing friendship and companionship.

Most participants reported keeping dogs primarily for security purposes. Dogs protected their households, livestock, and crops. One informant explained, "*A dog is both a pet and a source of security. It alerts you at night if there is any danger. When it barks, you go out and check. During the day, you can get an emergency travel, and your dog will guard against thieves or invaders. If invaders see that your dog is aggressive, they won't enter your compound. I would be uneasy without a dog since I cannot leave my homestead unattended.*" [Male Respondent]

Dogs protected their livestock and crops from wild predators such as mongoose, squirrels, and predatory birds like hawks and eagles. "*There was a time when I didn't have a dog, and mongoose would eat my chicken while I was asleep since there was no alert.*" [Female Respondent]

Other reasons for keeping dogs included hunting and herding, although these practices have diminished as the community has shifted to smallholder agriculture. "*It is mostly about taking care of the homestead now, and less about hunting or chasing away wild animals like cheetahs,*" [Female Respondent]. A few participants reported breeding dogs to sell puppies, while others gave them away to friends and neighbours for free.

When asked about the possibility of living without a dog, their response shifted from tangible to intangible roles such as friendship, companionship, societal norms, and status symbols. One participant remarked, "*This dog is like one of my kids. I love it. It has its own house and plate. It welcomes me when I get home,*" while another shared, "*It feels good to have a dog at home. It is not right to stay without a dog because others have them. We view dogs as flowers in a home.*" [Female Respondent]

We summarised the roles and values of dogs reported by community members as the following: security, societal norms, status symbols, herding, hunting, pet/friend, livestock and crop protection, breeding to sell puppies or give away to friends and other community members. These roles and values were further explored and ranked in the FGD.

**Pairwise ranking scores on the value of dogs from the FGD**

Two separate FGDs were held, one for female participants and another for male participants, each lasting approximately 2 hours. Tables 1 and 2 show the pairwise ranking scores for female and male FGD participants respectively. Female participants stressed the importance of a dog being a friend before it can be useful in any capacity. "*A dog that is not your friend cannot serve you,*" remarked one female participant. This is reflected in the equal ranking of security (home and livestock) roles and the companionship/friendship role by the female group.

Although the idea of a dog as a pet and friend ranked below their value for security (of the home, livestock, and crops in that order) among the men; they reiterated that security benefits depend on the foundational relationship of friendship with the dog. "*A dog that is not your friend cannot protect you, your livestock, nor your crops,*" a male participant remarked.

**1ˢᵗ co-creation workshop**

Two co-creation workshops were held with 24 community members. In the first workshop, participants emphasized the need for clear, visually engaging sensitization materials, preferring familiar bright colours like yellow. The rabies announcement poster should include the vaccination campaign's dates, time, and venue. Visuals showing veterinarians vaccinating dogs their owners restrain them will help build trust and depict an active vaccination process.

An educational poster should raise awareness of rabies symptoms, modes of transmission, prevention strategies, and actions to take in the event of a dog bite or scratch. *"We would like to see materials that are clear and self-explanatory,"* one participant shared.

The storyboarding process revealed community insights on the value of their dogs and how sensitization materials can be designed in a contextual manner. Participants illustrated their views on the roles of dogs in their lives, emphasizing security, companionship, and societal status. These storyboards captured the tangible benefits of dog ownership and highlighted the emotional and social bonds between the community members and their dogs (Fig 2).

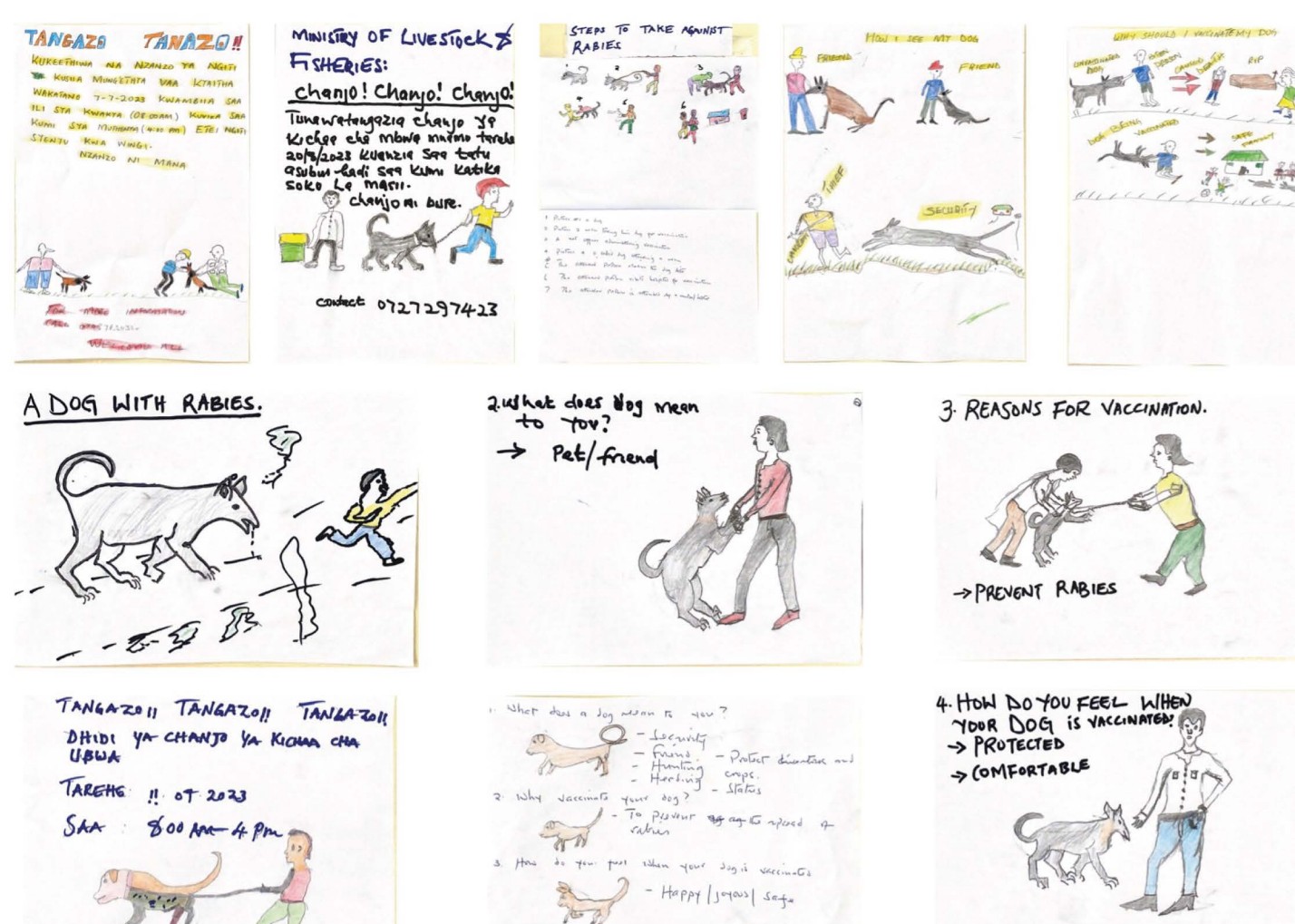

**Fig 2. Storyboard visuals developed by the participants illustrating their perceptions of dog values, motivations for vaccinating their dogs, and the value of the vaccination campaigns.** The visuals represent the elements to be highlighted in rabies sensitization materials. They also sketched out designs of rabies vaccination posters.

They shared that sensitization materials should be educational, culturally relevant and emotionally captivating to illustrate the human-dog bond.

*"Sketching and discussing our experiences has helped me understand that everyone value their dog differently. This is good because it brings us together and shows us new ways of protecting our community from rabies."* [Male FGD Participant].

**2nd co-creation workshop**

In the second co-creation workshop, participants reviewed three rabies vaccination announcement posters and one education poster. They emphasized visual clarity, cultural relevance, and simple design. They also stressed the use of bright

colours, clear messages, and local languages like Kikamba and Kiswahili to reach a wider audience. *"The bright colours speak to us. It's part of who we are as a Kamba community. Seeing these vibrant colours on the posters makes the message stand out. It draws attention and makes one desire to read and learn more about protecting our dogs and community from rabies,"* [Female Respondent].

The figures below present synthesized participants' feedback on the draft posters and images that guided the final design modifications.

Fig 3 was voted as the best announcement poster and was taken forward for final production. It had a clear layout and used bright colours that attracted attention. The font size was also legible, and the poster contained essential details such as the date, venue, and vaccination campaign costs.

Participants evaluated various images presented to them for their suitability based on dog values. Synthesized comments for each image are presented in Figs 3–5. Within Fig 6, images A and B were AI-generated while images C – F were captured during the vaccination campaigns.

There was a consensus to use images C and F found in Fig 6 in the rabies vaccination announcement poster. Picture C was selected because the vaccinator had personal protective equipment and demonstrated proper dog-handling techniques. Picture F illustrated a dog owner actively bringing their dogs to the vaccination site. It also highlighted gender equality and community involvement in rabies control programs.

"*The images in the sensitization materials should demonstrate proper practices. We would like to see that,*" [Male Respondent].

"*Seeing a true representation of our involvement in dog care by highlighting both men and women as active participants, reinforces the message of shared community responsibility,*" [Female Respondent].

The educational poster was praised for its organized layout, use of bright colours, and clear messaging. Suggestions included replacing the image of a tap with a water jug to reflect the local realities in rural settings. They also recommended changing the image of a doctor in a scrub suit to one in a white laboratory coat as they are accustomed to that. Additionally, participants suggested increasing the font size, and including an image of a dog owner restraining the dog while a veterinarian is vaccinating it (Fig 7).

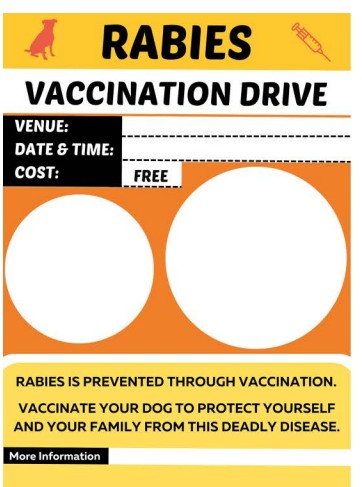

**Fig 3. Rabies announcement material (Poster 1).**

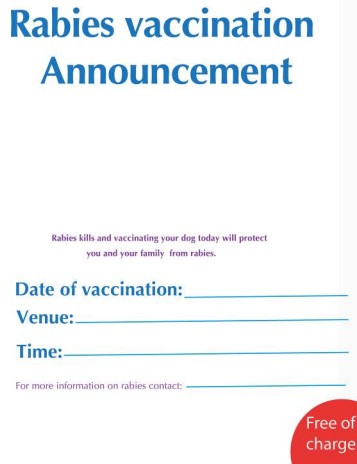

Poster 2 has a clear and starightforward design, effectively communicating esential details. However the lack of a bright colour makes it less visually striking. Suggestions for improvement, included increasing the font size to enhance legibility from a distance, removing the word "today" to facilitate better planning by the audience and adding translations in local languages to broaden its reach and accessibility.

**Fig 4. Rabies announcement material (Poster 2).**

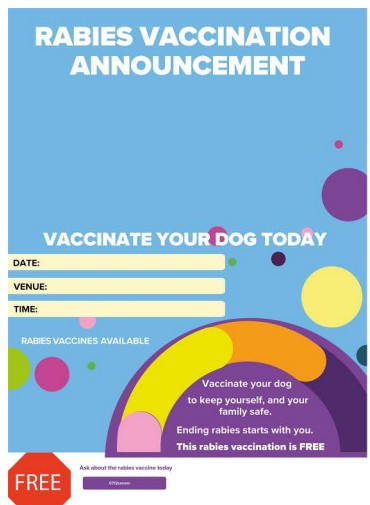

Poster 3 was recognized for its good messaging clarity. To further enhance its effectiveness, it was recommended to switch the blue color to a more vibrant hue to increase visibility, enhance the font size for improved legibility, and include an image of a dog being vaccinated to provide a practical visual cue that enriches the poster's informational value.

**Fig 5. Rabies announcement material (Poster 3).**

### Final co-created sensitization materials

The final sensitization materials incorporated participants' suggestions, the selected images, simple designs, and clear messages. Recommendations for bright colours and larger fonts were adopted for better visibility and legibility. Although posters with local language was encouraged, the final products were in English and will be translated at a later date (Fig 8 and Fig 9).

### Participants' perceptions of the participatory process, key learnings, and future wishes

Twenty-three out of 24 participants engaged in exit interviews, indicating a positive reception of the process. Participants primarily viewed the co-creation process as a form of training due to its emphasis on vaccinating dogs against rabies. They indicated that, through the process, they had learned what to do in the event of a dog bite.

PLOS Neglected Tropical Diseases

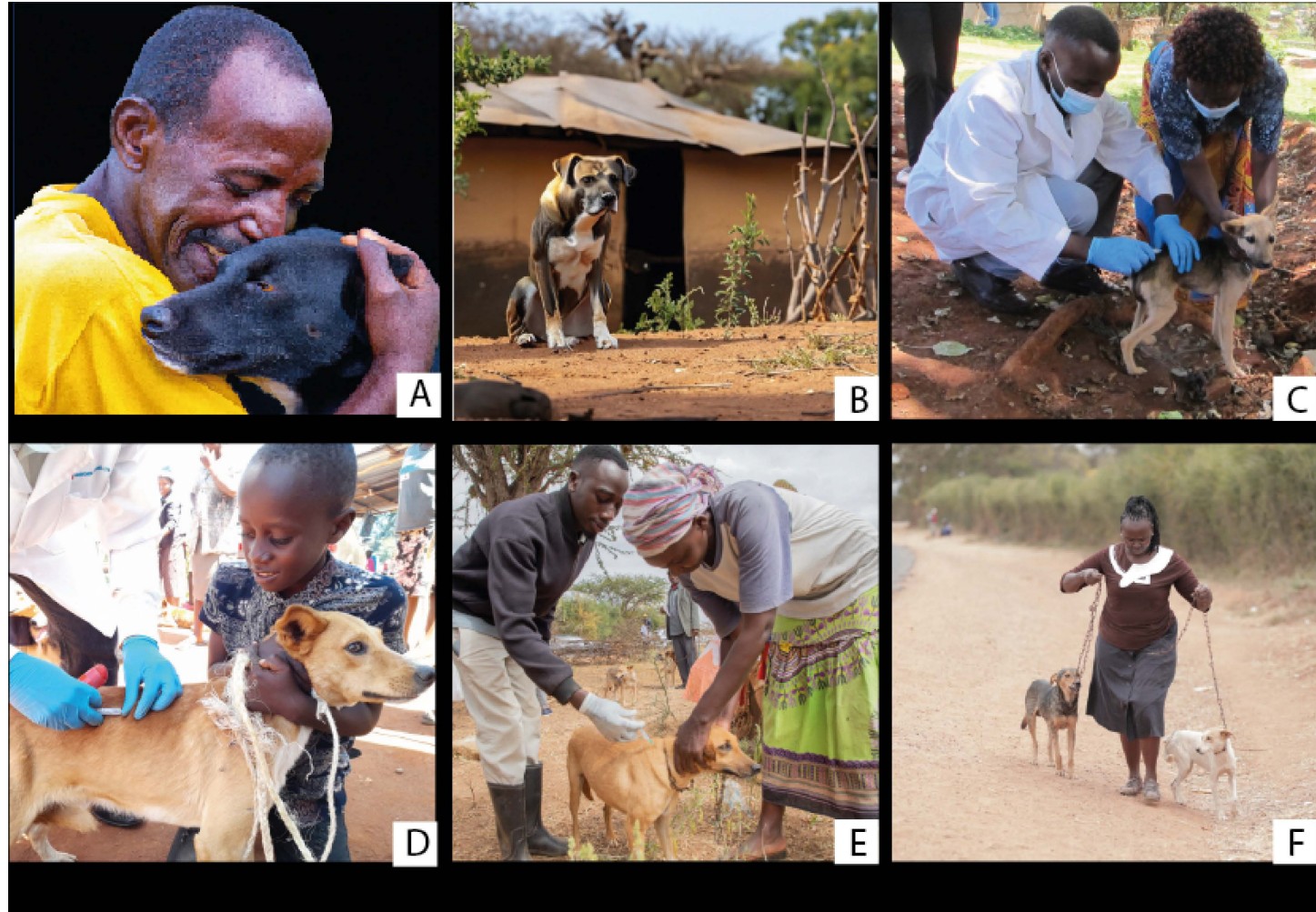

**Fig 6. Images presented and considered for inclusion in the sensitization materials.** Here are some of the synthesized comments for each image within Fig 6 A. They loved the human dog bond but suggested including a full picture of the person and the dog. B. Dog was healthy, however, the exotic breed shown is not commonly found in the study site. C. Showed proper vaccination procedure and handling technique. D. There was concerns about using a rope to restrain the dog which is a choking hazard and vaccinator is invisible. E. Concerns on the lack of personal protective equipment by the vaccinator. F. The lady taking her dogs showed an active process of community members presenting their dogs for vaccination.

"*I have learned about the importance of vaccinating dogs against rabies and taking bite victims to the hospital after a dog bite,*" [Respondent 12, Female].

"*I have learned that when bitten by a dog, one should wash the bite wound with soap and water before seeing the doctor. You should not apply onions on the bite wound as we have been doing. I have also learned that vaccinating your dog against rabies is important.*" [Respondent 6, Male].

Research participants mentioned that local animal health authorities should sensitize the community on rabies for better turn out in vaccination campaigns.

"*Continue sharing information about rabies with the community. Use media and distribute posters in schools, community meetings, and relevant gatherings.*" [Respondent 8, Female]

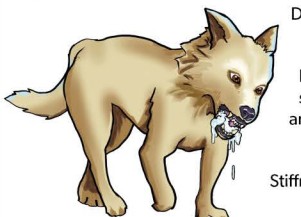
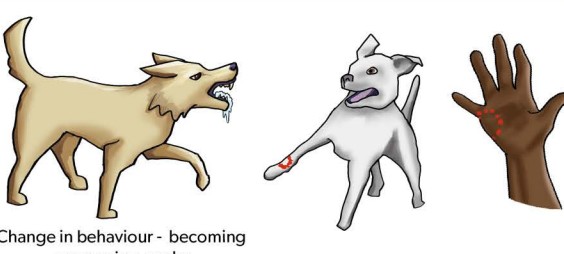
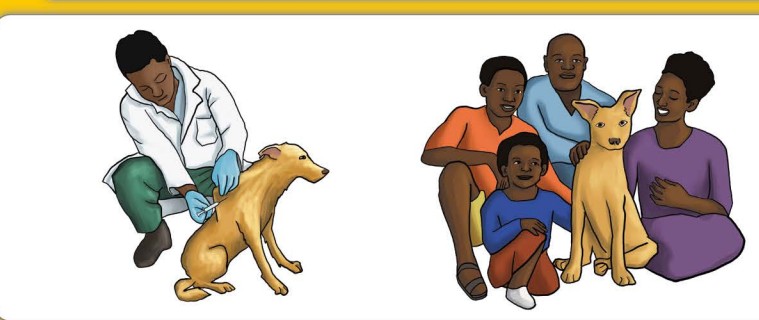
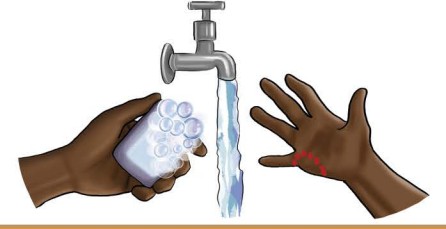
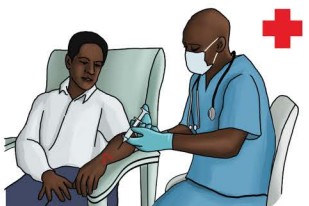

**Fig 7. Draft of the rabies educational poster.**

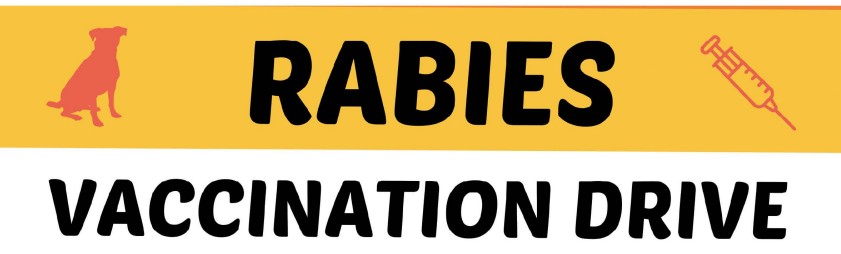

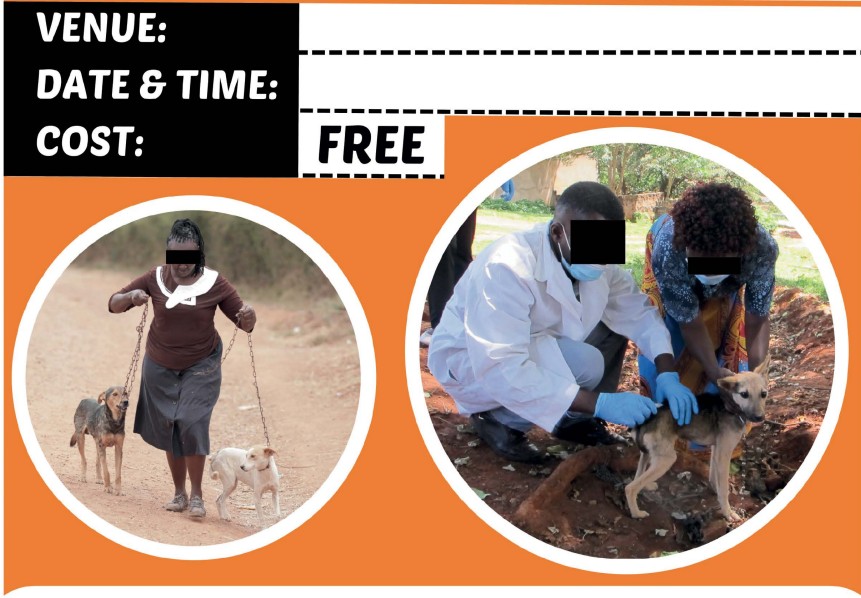

**Fig 8. Final rabies vaccination announcement poster.**

Regarding the co-creation process, participants felt their opinions were respected and considered by their peers.

"*I felt respected, listened to, and my opinions valued*," [Respondent 1, Male].

"*Drawing our dogs and seeing them on paper makes it real and important to keep them healthy and vaccinated*," [Female Participant].

# Why should you vaccinate your dog against rabies?

Rabies is 100% fatal for animals and humans, but it's also 100% preventable by vaccination.

Rabies is spread through bites from infected animals. If you don't vaccinate your dog, it could catch rabies, and infect other animals, and humans.

Signs of rabies include:

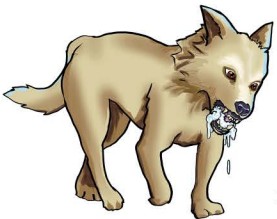

Distorted jaws

Excessive salivating and foaming

Stiffness in legs

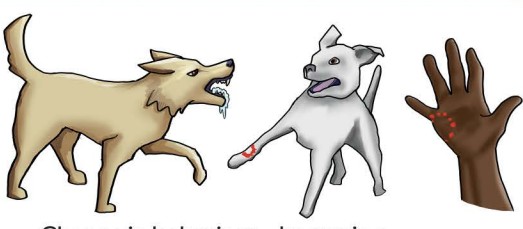

Change in behaviour - becoming aggressive, or shy

Vaccinate to keep your dog, yourself, and your family safe!

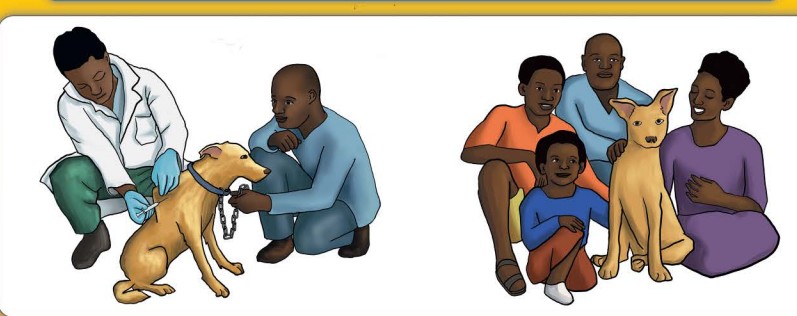

What to do when bitten by a dog:

1. Wash the bite wound with soap and water    2. Go to the hospital and see a doctor

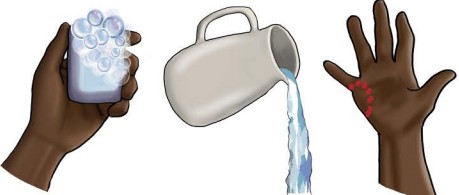
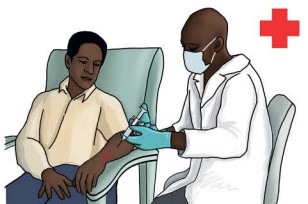

**Fig 9. Final rabies sensitization poster.**

## Discussion

This study explores the process, benefits and potential of incorporating community insights into the co-creation of rabies sensitization materials. As previously indicated, the co-created materials incorporated images and messages that emphasized the value of dogs. Human and animal health practitioners often advocate for the use of the One health approach in controlling rabies [1]. While this is a good starting point, engaging the community and raising awareness before conducting public health interventions may foster trust and drive behaviour change [23] which is critical for sustainability. Sustained control efforts are essential for diseases like rabies in order to adequately control or eliminate the disease in a population [24]. Involving the community in the planning, implementing, and monitoring stages of vaccination activities has been shown to be feasible with promising outcomes for mass dog vaccination [25].

Key barriers to achieving vaccination goals include low awareness of rabies and inadequate knowledge about vaccination campaigns, as observed during a rabies outbreak in Peru [9]. Similarly, the lack of rabies educational programs was a hindrance to achieving 70% vaccination coverage in Laikipia County in Kenya. In these campaigns, they achieved a vaccination coverage of between 2 – 24% over the 3-year period [26] which is below the recommended 70% threshold.

The co-creation process allowed dog owners to reflect on the value of their dogs and recognized the importance of vaccinating them whom they described as their 'friends.' Another key finding was the desire for simple and clear messaging in the sensitization materials. This aligns with findings from another study which recommended that health messages should be memorable and actionable to the target audience [27]. There was the desire to disseminate key messages to other community members. We hope the co-creation process established a cohort of community champions equipped with accurate information on rabies control. These findings resonate with studies in Tanzania, Kenya, and Burkina Faso that established that rabies sensitization increased the likelihood of rabies vaccine uptake [11,26,28]. The effective communication of key messages and planning for vaccination should involve the community and reflect local realities [29]. Improving the dog's owners knowledge of rabies has been established to increase the intention of vaccinating their dogs [30] hence the relevance of these sensitization materials. These materials also incorporated visual aids (images) which have been found to be missing in rabies leaflets [31].

Participatory research has been shown to improve study outcomes, data accessibility, and public transparency in environmental health [32]. Our findings support the use of participatory and non-fear-driven approaches which may lead to successful health outcomes. It is worth noting that community engagement -which is perceived as a bottom-up approach - tends to be more expensive but more sustainable than a top-down approach [23,33]. Sanitation interventions with minimal community involvement during the planning and implementation stages were unsuccessful in India. This failure is attributed to a lack of local ownership, contextual relevance, and sustained behavior change. Excluding communities from decision-making processes often results in interventions that are misaligned with local cultural practices, and user preferences, thereby leading to poor adoption, limited use, and inadequate maintenance of sanitation infrastructure. [34]. Cost-effective analysis is important, especially in resource-limited settings as it is cheaper to vaccinate dogs against rabies than purchase post exposure prophylaxis for victims.

Our work builds on the recommendations by Murungi et al [35] that future rabies vaccination campaigns be accompanied with increased community engagement.

The key limitation of our study is that research participants were selected from community members who presented their dogs at vaccination sites. They are therefore more likely to reflect the perceptions of those community members already interested and engaged in caring for their dogs and controlling rabies. The other limitation is that the interview guide was drafted in English and then back-translated to Kiswahili and Kikamba. However, trained multilingual enumerators ensured conceptual consistency so any slight wording differences are unlikely to have introduced meaningful subjective bias. Future research should evaluate the impact of incorporating community sensitization on rabies control by comparing vaccination uptake and coverage in areas where sensitization is conducted versus the sites where sensitization is not conducted.

**Neglected Tropical Diseases**

## Conclusion

This study demonstrated that dogs in Machakos County hold both tangible and intangible value for their owners—providing security for homes, livestock, and crops, as well as companionship and friendship, which was seen as a prerequisite for meaningful service. By engaging communities through key informant interviews, focus group discussions, and co-creation workshops, we uncovered how these values may positively shape rabies control practices. The participatory co-creation process not only raised rabies awareness but also encouraged reflection on the role of dogs and motivated owners to vaccinate them.

These findings highlight that integrating community insights into health interventions can improve uptake by aligning communication with local values and lived experiences. We recommend that rabies control programs move beyond top-down messaging and instead co-create educational and promotional materials with communities to foster trust, increase awareness, and enhance participation. This approach is essential for achieving sustainable vaccination coverage and should be adopted for other One Health initiatives targeting diseases of public health importance.

## Acknowledgments

We thank officials from the Ministry of Agriculture, Food Security, and Cooperative Development, Machakos County for their support in planning this study. We acknowledge Madeline Wong, Fenja Tramsen, and Kristen Tam for capturing the photos during the co-creation workshops. Special acknowledgment to all the twenty-four participants who co-created the sensitization materials and consented to being named on outputs from this project. They include Ann Kiseli, Ann Ngei, Carolyne Mutie, Catherine Mueni, Daniel Muli, Elizabeth Musyoka, Esther Mutinda, Felix Mwangangi, Henry Makenzi, Joel Kioko, John Wambua, Joseph Mutisya, Josephine Mutua, Nelson Mandela, Nicholas Ndolo, Patrick Kimeu, Patrick Muthiani, Phyllis Kithama, Rebecca Kioko, Rebecca Mumbe, Rhoda Peter, and Ruth Mumba.

## Author contributions

**Conceptualization:** Nicholas Bor, Lian F. Thomas.

**Data curation:** Nicholas Bor, Peterkin Munywoki, Lian F. Thomas.

**Formal analysis:** Nicholas Bor, Lian F. Thomas.

**Funding acquisition:** Lian F. Thomas.

**Investigation:** Nicholas Bor, Geoffrey Njenga, Peterkin Munywoki, Dorcas Chepyatich.

**Methodology:** Nicholas Bor, David Owino, Lian F. Thomas.

**Project administration:** Nicholas Bor, Lian F. Thomas.

**Software:** Geoffrey Njenga.

**Supervision:** Lian F. Thomas.

**Visualization:** Geoffrey Njenga, Annabel Slater.

**Writing – original draft:** Nicholas Bor, Geoffrey Njenga, Lian F. Thomas.

**Writing – review & editing:** Nicholas Bor, Geoffrey Njenga, Annabel Slater, Peterkin Munywoki, Dorcas Chepyatich, David Owino, Maurice K. Murungi, Dishon M. Muloi, Lian F. Thomas.

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
