## [Decision Letter · Decision Letter 0]

8 May 2025

PNTD-D-25-00146Rabies Control via Co-Creation: A Model for Sustainable One Health InterventionsPLOS Neglected Tropical Diseases  Dear Dr. Bor, Thank you for submitting your manuscript to PLOS Neglected Tropical Diseases. After careful consideration, we feel that it has merit but does not fully meet PLOS Neglected Tropical Diseases's publication criteria as it currently stands. Therefore, we invite you to submit a revised version of the manuscript that addresses the points raised during the review process. Please submit your revised manuscript within 30 days Jul 07 2025 11:59PM. If you will need more time than this to complete your revisions, please reply to this message or contact the journal office at plosntds@plos.org. Please include the following items when submitting your revised manuscript: * A rebuttal letter that responds to each point raised by the editor and reviewer(s). You should upload this letter as a separate file labeled 'Response to Reviewers '. This file does not need to include responses to any formatting updates and technical items listed in the 'Journal Requirements' section below. * A marked-up copy of your manuscript that highlights changes made to the original version. You should upload this as a separate file labeled 'Revised Manuscript with Track Changes '. * An unmarked version of your revised paper without tracked changes. You should upload this as a separate file labeled 'Manuscript '. If you would like to make changes to your financial disclosure, competing interests statement, or data availability statement, please make these updates within the submission form at the time of resubmission. Guidelines for resubmitting your figure files are available below the reviewer comments at the end of this letter. We look forward to receiving your revised manuscript. Kind regards, Richard A. Bowen, DVM PhDAcademic EditorPLOS Neglected Tropical Diseases Victoria BrookesSection EditorPLOS Neglected Tropical Diseases

Shaden Kamhawi

co-Editor-in-Chief

Paul Brindley

co-Editor-in-Chief

**Additional Editor Comments:** Your manuscript has been reviewed by 3 experts, each of whom applauded your effort and has provided major or minor suggestions for clarification or improvement. Please review these comments, revise your manuiscript as you think best and resubmit.**Journal Requirements:**

At this stage, the following Authors/Authors require contributions: Nicholas Bor, Geoffrey Njenga, Annabel Slater, Peterkin Munywoki, Dorcas Chepyatich, David Owino, Maurice K. Murungi, Dishon Muloi, and Lian F. Thomas. Please ensure that the full contributions of each author are acknowledged in the "Add/Edit/Remove Authors" section of our submission form.

Potential Copyright Issues:

- Please confirm (a) that you are the photographer of Figures 1, 2, 6, and 8, or (b) provide written permission from the photographer to publish the photo(s) under our CC BY 4.0 license.

- Figures 3, 4, 5, 7, and 8. Please confirm whether you drew the images / clip-art within the figure panels by hand. If you did not draw the images, please provide (a) a link to the source of the images or icons and their license / terms of use; or (b) written permission from the copyright holder to publish the images or icons under our CC BY 4.0 license. Alternatively, you may replace the images with open source alternatives. See these open source resources you may use to replace images / clip-art:

- The following Figures contain a logo or branding: Figures 3, 4, 5, 7, and 8. We are not permitted to publish this under our CC-BY 4.0 license, even with permission. We ask that you please remove or replace it.

5) We note that your Data Availability Statement is currently as follows: "N/A". Please confirm at this time whether or not your submission contains all raw data required to replicate the results of your study. Authors must share the “minimal data set” for their submission. PLOS defines the minimal data set to consist of the data required to replicate all study findings reported in the article, as well as related metadata and methods (https://journals.plos.org/plosone/s/data-availability#loc-minimal-data-set-definition).

- The points extracted from images for analysis..

6) Please amend your detailed Financial Disclosure statement. This is published with the article. It must therefore be completed in full sentences and contain the exact wording you wish to be published. Please ensure that the funders and grant numbers match between the Financial Disclosure field and the Funding Information tab in your submission form. Note that the funders must be provided in the same order in both places as well.

**Reviewers' comments:** Reviewer's Responses to Questions

**Key Review Criteria Required for Acceptance?**

**Methods:**

-Are the objectives of the study clearly articulated with a clear testable hypothesis stated?

-Is the study design appropriate to address the stated objectives?

-Is the population clearly described and appropriate for the hypothesis being tested?

-Is the sample size sufficient to ensure adequate power to address the hypothesis being tested?

-Were correct statistical analysis used to support conclusions?

-Are there concerns about ethical or regulatory requirements being met?

Reviewer #1: Please see attached comments.

Reviewer #2: (No Response)

Reviewer #3: Yes, to all of the above questions. MH

**Results**

-Does the analysis presented match the analysis plan?

-Are the results clearly and completely presented?

-Are the figures (Tables, Images) of sufficient quality for clarity?

Reviewer #1: See attached comments.

Reviewer #2: (No Response)

Reviewer #3: Yes, to all of the above questions. MH

**Conclusions**

-Are the conclusions supported by the data presented?

-Are the limitations of analysis clearly described?

-Do the authors discuss how these data can be helpful to advance our understanding of the topic under study?

-Is public health relevance addressed?

Reviewer #1: See attached comments.

Reviewer #2: (No Response)

Reviewer #3: Yes, to all of the above questions. This is a very novel idea for the bottom-up approach to public health. MH

**Editorial and Data Presentation Modifications?**

Reviewer #1: See attached comments.

Reviewer #2: (No Response)

Reviewer #3: N/A

**Summary and General Comments**

Reviewer #1: This manuscript presents findings from a co-creation strategy aimed at developing communication materials for rabies awareness and mass dog vaccination campaigns (MDVC). The study makes a valuable contribution to the rabies literature by addressing a critical, yet often overlooked, barrier to successful MDVC implementation—namely, communication challenges and insufficient community engagement. By involving community stakeholders in the development of communication tools, the study offers a practical and context-sensitive approach to improving public health interventions. I recommend this manuscript for publication, pending minor revisions that I believe will strengthen the clarity and overall impact of the work.

Reviewer #2: (No Response)

Reviewer #3: To the author, I am not sure who or how this idea was developed, but I really like it and wish I had thought of something similar with my own rabies work. Community work is fulfilling, and I am sure you feel that something good was accomplished here based upon all of the comments the participants provided. Thank you for including those in the manuscript. More effective and believable to read their actual comments. This community buy in and participation is exciting and I hope that others will model the idea. Thank you. I enjoyed reading and reviewing your manuscript.

PLOS authors have the option to publish the peer review history of their article (what does this mean? ). If published, this will include your full peer review and any attached files.

**Do you want your identity to be public for this peer review?** For information about this choice, including consent withdrawal, please see our Privacy Policy .

Reviewer #1: No

Reviewer #2: No

Reviewer #3: **Yes: ** Melinda Hergert

---

## [Editor Report · Decision Letter 1]

11 Jul 2025

Dear Bor,

We are pleased to inform you that your manuscript 'Rabies Control via Co-Creation: A Model for Sustainable One Health Interventions' has been provisionally accepted for publication in PLOS Neglected Tropical Diseases.

Best regards,

Richard A. Bowen, DVM PhD

Academic Editor

Victoria Brookes

Section Editor

Shaden Kamhawi

co-Editor-in-Chief

Paul Brindley

co-Editor-in-Chief

Thank you for addressing the comments and suggestions from reviewers. I believe this is a valuable addition to the highly important quest to promote vaccination of dogs against rabies to decrease the burden of human rabies. I especially enjoyed all of the posters and other graphics you provided.

---

## [Editor Report · Acceptance letter]

Dear Mr Bor,

We are delighted to inform you that your manuscript, "Rabies Control via Co-Creation: A Model for Sustainable One Health Interventions," has been formally accepted for publication in PLOS Neglected Tropical Diseases.

Best regards,

Shaden Kamhawi

co-Editor-in-Chief

Paul Brindley

co-Editor-in-Chief
